# Effect of Flank Rotation on the Photovoltaic Properties of Dithieno[2,3-*d*:2′,3′-*d*′]benzo[1,2-*b*:4,5-*b*′]dithiophene-Based Narrow Band Gap Copolymers

**DOI:** 10.3390/polym11020239

**Published:** 2019-02-01

**Authors:** Mingjing Zhang, Liangjian Zhu, Pengzhi Guo, Xunchang Wang, Junfeng Tong, Xiaofang Zhang, Yongjian Jia, Renqiang Yang, Yangjun Xia, Chenglong Wang

**Affiliations:** 1School of Materials Science and Engineering, Lanzhou Jiaotong University, Lanzhou 730070, China; xijian819mj@163.com (M.Z.); tongjunfeng139@163.com (J.T.); 13008709729@163.com (X.Z.); 2Key Laboratory of Optoelectronic Technology and Intelligent Control of Ministry Education, Lanzhou Jiaotong University, Lanzhou 730070, China; phljzhu@sina.com; 3National Green Coating Technology and Equipment Engineering Technology Research Center, Lanzhou Jiaotong University, Lanzhou 730070, China; shxygpz@126.com (P.G.); jiayj88@hotmail.com (Y.J.); 4CAS Key Laboratory of Bio-Based Materials, Qingdao Institute of Bioenergy and Bioprocess Technology, Chinese Academy of Sciences, Qingdao 266101, China; wang_xc@qibebt.ac.cn (X.W.); yangrq@qibebt.ac.cn (R.Y.)

**Keywords:** dithieno[2,3-*d*:2′,3′-*d*′]benzo[1,2-*b*:4,5-*b*′]dithiophene, 4,5-didecylthien-2-yl-ethynyl, 4,5-didecylthien-2-yl, exciton dissociation probability, time-resolved photoluminescence

## Abstract

Side chain engineering has been an effective approach to modulate the solution processability, optoelectronic properties and miscibility of conjugated polymers (CPs) for organic/polymeric photovoltaic cells (PVCs). As compared with the most commonly used method of introducing alkyl chains, the employment of alkyl-substituted aryl flanks would provide two-dimensional (2-D) CPs having solution processability alongside additional merits like deepened highest occupied molecular orbital (HOMO) energy levels, increased absorption coefficient and charger transporting, etc. In this paper, the triple C≡C bond was used as conjugated linker to decrease the steric hindrance between the flanks of 4,5-didecylthien-2-yl (T) and dithieno[2,3-*d*:2′,3′-*d*′]benzo[1,2-*b*:4,5-*b*′]dithiophene (DTBDT) core. In addition, an alternating CP derived from 4,5-didecylthien-2-yl-ethynyl (TE) flanked DTBDT, and 4,9-bis(4-octylthien-2-yl) naphtho[1,2-*c*:5,6-*c*′]bis[1,2,5]thiadiazole (DTNT), named as PDTBDT-TE-DTNT, was synthesized and characterized. As compared with the controlled PDTBDT-T-DTNT, which was derived from 4,5-didecylthien-2-yl flanked DTBDT and DTNT, the results for exciton dissociation probability, density functional theory (DFT), time-resolved photoluminescence (PL) measurements, etc., revealed that the lower steric hindrance between TE and DTBDT might lead to the easier rotation of the TE flanks, thus contributing to the decrease of the exciton lifetime and dissociation probability, finally suppressing the short-circuit current density (*J*_SC_), etc., of the photovoltaic devices from PDTBDT-TE-DTNT.

## 1. Introduction

Polymeric photovoltaic cells (PVCs) have attracted much attention due to advantages such as the printable fabrication process, and their ultrathin, lightweight and flexible properties. These unique characteristics of the PVCs could provide more favorable prospects for upcoming power generating windows, automobiles, wearable electronics, etc., in contrast to their inorganic counterparts [1,2,3,4]. The printable fabrication process and flexibility of the PVCs are mainly attributed to the solution processability and flexibility of the conjugated polymers (CPs), which are strongly related with their side chains. As compared with the most common way, whereby alkyl side chains are used to develop soluble CPs [5,6,7,8], the employment of alkyl-substituted aryl side chains to build up the two-dimensional (2-D) CPs not only provides modulation of the solubility and aggregation, but also endows the corresponding 2-D CPs with the merits of additionally decreasing the highest occupied molecular orbital (HOMO) energy levels, increasing the absorption coefficient, modifying the charger transport, etc. [9,10,11,12]. In recent years, various 2-D CPs have been developed. In addition, the 2-D CPs from alkyl-substituted aryl side chains flanking benzo[1,2-*b*:4,5-*b*′]dithiophene (BDT) derivatives have proved to be the most promising CPs for highly efficient PVCs [10,11,12,13,14,15,16]. For instance, Hou, Li, Huo and Lee et al. employed alkyl-substituted thien-2-yl to function as BDT, and the 2-D donor–acceptor (D–A) CPs deriving from the alkylthiophene-substituted BDT with alkoxycarbonyl-substituted thieno[3,4-*b*]thiophene (TT-E) [13], alkylcarbonyl-substituted thieno[3,4-*b*]thiophene (TT-C) [13], quinoxaline derivatives (DTQx) [14], 1,3-bis(thioph-en-2-yl)-5,7-bis(2-ethylhexyl)benzo[1,2-*c*:4,5-*c*′]dithiophene-4,8-dione (BDD) [15], dialkoxybenzo-thiadiazole [16], etc., were built up. They demonstrated that the 2-D CPs exhibited better thermal stabilities, red-shifted absorption spectra, lower HOMO energy levels, significantly enhanced hole mobility, etc., as compared to their counterpart CPs with alkyl side chains. Additionally, alkyl-substituted aryl side chains such as alkylthiothien-2-yl [17], alkylfuryl [18], alkylselenophen-2-yl [18], alkylphenyl [19], alkylthieno[3,2-*b*]thiophen-2-yl [20], etc., are widely implemented to develop novel BDT-based 2-D CPs.

Dithieno[2,3-*d*:2′,3′-*d*′]benzo[1,2-*b*:4,5-*b*′]dithiophene (DTBDT), an aromatic analogue of BDT, not only shows similar HOMO levels with BDT, but also has a larger coplanar core and an extended conjugation length. It is believed that DTBDT-based CPs could provide some advantageous properties such as enhanced charge-carrier mobility, decreased band gaps and promotion of exciton separation into free charge carriers in contrast to BDT-based CPs [21,22,23,24,25,26,27,28]. Since Hou et al. reported a 2-D CP (PDT-S-T) based on 5,10-di(5-(2-ethylhexylthien-2-yl)-DTBDT, and a PCE of 7.79% was achieved in the PVCs from PDT-S-T in 2013 [22]. Many DTBDT-based 2-D CPs have been developed. For example, we presented a polymer (PBT-T-DPP-C12) with the alkylthien-2-yl flanked DTBDT as the electron donor moieties and diketopyrrolopyrrole (DPP) as the electron acceptor moieties in 2014 [23]. Following that, Huo, Huang and We et al., presented several high-performance 2-D CPs with alkylthiophene-flanked DTBDT as electron donor units, and 1,3-bis(5-bromothiophen-2-yl)-5,7- bis(2-ethylhexyl)-4*H*,8*H*-benzo[1,2-*c*:4,5-*c*′]dithiophene-4,8-dione [24], 4,9-di(4-hexylthien-2-yl)naph- tha[1,2-*c*:5,6-*c*′]bis[1,2,5]thiadiazole [25], 4,8-di(thien-2-yl)-6-octyl-2-octyl-5*H*-pyrrolo[3,4-*f*]benzotri- azole-5,7(6*H*)-dione [26], benzothiadiazole, 5,6-difluoro-benzothiadiazole [27], etc., as electron acceptor units. The advantages, such as deepened HOMO levels, broadened light response, enhanced charge transporting, etc., of the 2-D DTBDT-based CPs, were presented in comparison with their counterpart CPs from alkyl-substituted DTBDT [24,25,26,27]. Most recently, we provided two alternating medium-band-gap CPs (PBDT-TPTI and PDTBDT-TPTI) derived from alkylthiophene-substituted BDT (BDT-T) or alkylthiophene-substituted DTBDT (DTBDT-T) and *N*,*N*- didodecylthieno[2ʹ,3ʹ:5,6]pyrido[3,4-*g*]thieno[3,2-*c*]-*iso*-quinoline-5,11-dione (TPTI). Comparative investigations of the CPs revealed that the optimized geometries of the models of BDT-T and DTBDT-T presented the dihedral angles of 75.5° and 52.5° between the flank of 4,5-dialkylthien-2-yl and DTBDT and/or BDT core, and which played an important role in affecting the absorption, aggregation characteristics and optoelectronic properties of the CPs [28]. Meanwhile, it has been well demonstrated that the enhanced steric hindrance between flanks and conjugated backbone would result an increase in the dihedral angles between the flanks and backbone of CPs, thus leading to decreased coplanarity, and adjusting their aggregation, optoelectronic properties [18,28].

In this paper, the triple C≡C bond was used as a conjugated linker to decrease the steric hindrance between the flanks of 4,5-didecylthien-2-yl (T) and dithieno[2,3-*d*:2′,3′-*d*′]benzo[1,2-*b*: 4,5-*b*′]dithiophene (DTBDT) core. In addition, the alternating narrow band gap CP named PDTBDT- TE-DTNT, derived from 5,10-bis(4,5-didecylthien-2-yl-ethynyl)dithieno[2,3-*d*:2′,3′-*d*′]benzo[1,2-*b*: 4,5-*b*′]dithiophene (DTBDT-TE) and 4,9-di(4-octylthien-2-yl)naphtho[1,2-*c*:5,6-*c*′]bis[1,2,5]thiadia- zole (DTNT), was synthesized and characterized. Following that, comparative investigation of the optoelectronic and aggregation properties between PDTBDT-TE-DTNT and the controlled copolymer of poly{(5,10-bis(4,5-didecylthien-2-yl)dithieno[2,3-*d*:2′,3′-*d*′]benzo[1,2-*b*:4,5-*b*′]dithiopene-2,7-*alt*-4,9- di(4-octylthien-2-yl)naphtho[1,2-*c*:5,6-*c*′]bis[1,2,5]thiadiazole5,5’-yl)} (PDTBDT-T-DTNT), in which the alkyl-substituted aryl side chain of T was directly linked to DTBDT, were implemented by the UV-Vis absorption, temperature-dependent absorption spectra (TD-Abs), X-ray diffraction (XRD), cyclic voltammetry (CV), etc., measurements. In addition, their optimal inverted photovoltaic devices (*i-*PVCs) provided power conversion efficiencies (PCEs) of 3.97% and 7.57%, with open circuit voltages (*V*_OC_) of 0.60 and 0.70 V, short-circuit current densities (*J*_SC_) of 10.15 and 16.09 mA/cm^2^ and fill factors (*FF*) of 65.16% and 67.19% under 100 mW/cm^2^ illumination (AM 1.5G), respectively. It has been demonstrated that the replacement T with 4,5-didecylthien-2-yl-ethynyl (TE) flanks would lead to the slightly broadening of the band gap, elevating the HOMO energy levels of the PDTBDT-TE-DTNT relative with PDTBDT-T-DTNT. Moreover, the optical calculations, exciton dissociation probability, time-resolved photoluminescence (PL) measurements, computational considerations, etc., revealed that the lower torsional barrier of the TE flanks, might contribute to the enhancement of the relaxation patterns of the excitons in the PDTBDT-TE-DTNT, leading to a decrease in the photo-induced excited state lifetime and the exciton dissociation probability, which were mainly attributed to the lower *J*_SC_ of the PVCs from PDTBDT-TE-DTNT, as compared with those for PDTBDT-T-DTNT- and PDTBDT-T-DTNT-based PVCs.

## 2. Materials and Methods

All reagents, unless otherwise specified, were obtained from Alading (Shanghai, China), Acros (Bridgewater, NJ, USA) and TCI Chemical Co. (Shanghai, China), and used as received. 4,5-didecylthiophene [23], dithieno[2,3-*d*:2′,3′-*d*′]benzo[1,2-*b*:4,5-*b*′]dithio- phene-5,10-diketone [23], 2,7-di(trimethylstannyl)-5,10-bis(4,5-didecylthien-2-yl)dithieno[2,3-*d*:2′,3′-*d′*]benzo[1,2-*b*:4,5-*b′*]dithiophene (DTBDT-TSn) [23], 4,9-di(5-bromo-4-octylthien-2-yl)naphtha[1,2-*c*: 5,6-*c′*]bis[1,2,5]thiadiazole (DTNTC_8_Br) [25], and poly[(9,9-bis(3′-(*N*,*N*-dimethylamino)propyl)-2,7-fluorene)-*alt*-2,7-(9,9-dioctylfluorene)] (PFN) [29,30] were synthesized by the procedure reported in the reference, and characterized by ^1^H NMR before use. Tetrahydrofuran (THF) and toluene were dried by sodium with benzophenone as indicator under a nitrogen flow. The molecular weights, absorption, electrochemical, aggregation characteristics were characterized following our previous methods in the reference and presented in Appendix A.

### 2.1. Synthesis of the Monomers and Copolymer

#### 2.1.1. 2-iodo-4,5-didecylthiophene (T-I)

4,5-Didecylthiophene (2.00 g, 5.49 mmol) was dissolved in chloroform (35 mL) and HOAc (7 mL) under argon at 0 °C, followed by slow addition of *N*-iodosuccinimide (NIS) (1.73 g, 7.68 mmol) under dark. The mixture was stirred for 20 min and allowed to warm to room temperature (r.t.). After stirring for 4 h, dilute aqueous Na_2_S_2_O_3_ (20 mL) was added to the solution then the solution was extracted with chloroform. The combined organic layer was dried over anhydrous Na_2_SO_4_ and concentrated. Purification by silica gel chromatography using n-hexane gave white oil. The product was obtained to 2.30 g (yield: 85.2%).^1^H NMR (500 MHz, CDCl_3_) δ (ppm): 7.28 (s, 1H), 2.80–2.77 (m, 2H), 2.62–2.53 (m, 2H), 1.72–1.59 (m, 4H), 1.42–1.28 (m, 28H), 0.88 (m, 6H).

#### 2.1.2. 2-trimethylsilylethynyl-4,5-didecylthiophene (TE-Si)

T-I (1.00 g, 2.04 mmol), palladium chloride (28.95 mg, 0.16 mmol), cuprous iodide (24.83 mg, 0.13 mmol), triphenylphosphine (0.21 g, 0.80 mmol) were diluted with triethylamine (5 mL) and THF (5 mL) under nitrogen, 2-trimethylsilyl-acetylene (0.28 g, 2.86 mmol) was slowly added to the mixture. After all of the solution had been added, the mixture was heated at 65 °C with stirring for 1 h. It was then poured into water (300 mL) and extracted with ethyl acetate, the organic phase was washed with brine (200 mL), dried with anhydrous Na_2_SO_4_ and concentrated. The crude product was purified by column chromatography (hexane eluent) to obtain 0.73 g of clear oil (yield: 75.0%). ^1^H NMR (500 MHz, CDCl_3_) δ (ppm): 7.30 (s, 1H), 2.80–2.77 (m, 2H), 2.62–2.53 (m, 2H), 1.72–1.59 (m, 4H), 1.42–1.28 (m, 28H), 0.88 (m, 6H), 0.08 (m, 9H).

#### 2.1.3. 2-(4,5-didecylthien-2-yl)acetylene (TE)

TE-Si (2.00 g, 4.35 mmol)) was dissolved in THF (30 mL) to which a solution of 3.00 g of potassium hydroxide dissolved in 30 mL of methanol was added at r.t. The mixture was stirred for 1 h then add 200 mL of water and extracted with CH_2_Cl_2_ (200 mL). The organic phase was dried with anhydrous Na_2_SO_4_ and concentrated. The crude product was purified by column (hexane) to obtain 1.65 g of oil (yield: 98.0%). ^1^H NMR (500 MHz, CDCl_3_) δ (ppm): 7.29 (s, 1H), 4.04 (s, 1H), 2.80–2.77 (m, 2H), 2.62–2.53 (m, 2H), 1.72–1.59 (m, 4H), 1.42–1.28 (m, 28H), 0.88 (m, 6H).

#### 2.1.4. 5,10-di(4,5-didecylthien-2-yl-ethynyl)dithieno[2,3-*d*:2′,3′-*d*′]benzo[1,2-*b*:4,5-*b*′]dithiophene (DTBDT-TE)

To a solution of TE (1.75 g, 4.50 mmol) in anhydrous THF (30 mL) at −10 °C, n-BuLi (2.5 M in hexane, 2.8 mL, 4.50 mmol) was added dropwise. The reaction mixture was stirred for 2 h then heated to 50 °C for 2 h. Subsequently, dithieno[2,3-*d*:2′,3′-*d*′]benzo[1,2-*b*:4,5-*b*′]dithiophene-5,10- diketone (0.50 g, 1.50 mmol) was quickly added and the mixture was stirred at 50 °C for 1 h. Then a solution of SnCl_2_·2H_2_O (2.70 g, 12.00 mmol) in 10% HCl (30 mL) was added and stirred for additional 1.5 h and then poured into the ice water. The mixture was extracted twice with petroleum ether (PE). The organic phase was dried over Na_2_SO_4_ and concentrated to afford crude product, which was purified by silica gel chromatography using PE as the eluent to yield a yellow solid (yield: 1.46 g, 91.3%). ^1^H NMR (500 MHz, CDCl_3_) δ (ppm): 7.60 (d, *J* = 6.0 Hz, 1H), 7.41 (d, *J* = 6.0 Hz, 1H), 7.30 (s, 1H), 2.80–2.77 (m, 2H), 2.56–2.53 (m, 2H), 1.72–1.59 (m, 4H), 1.42–1.29 (m, 28H), 0.88 (m, 6H).

#### 2.1.5. 2,7-di(trimethylstannyl)-5,10-di(4,5-didecylthien-2-yl-ethynyl)dithieno[2,3-*d*:2′,3′-*d*′]benzo[1,2-*b*:4,5-*b*′]dithiophene (DTBDT-TESn)

Under nitrogen at −35 °C, *n*-BuLi (1.6 M in hexane, 0.86 mL, 1.38 mmol) was added dropwise to the 60 mL of anhydrous THF solution of DTBDT-TE (0.50 g, 0.46 mmol), the reaction mixture was stirred for 2 h. Then, at −35 °C, hexane solution of Me_3_SnCl (0.29 g, 1.47 mmol) was added in one portion. The reaction mixture was stirred at −35 °C for 0.5 h and then warmed to r.t. for 6 h. Subsequently, the reaction mixture was poured into water and extracted twice with PE. Then, the organic layer was dried over Na_2_SO_4_ and concentrated to obtain the yellow crude product, which was recrystallized from *iso*-propanol (*i-*PrOH), finally obtaining pure DTBDT-TE as a yellow sheet-shaped crystal (yield: 0.55 g, 85.0%). ^1^H NMR (500 MHz, CDCl_3_) δ (ppm): 7.41 (s, 1H), 7.31 (s, 1H), 2.81–2.77 (m, 2H), 2.56–2.53 (m, 2H), 1.72–1.59 (m, 4H), 1.42–1.28 (m, 31H), 0.89 (m, 6H), 0.53–0.42 (m, 9H). (Appendix A). ^13^C NMR (125 MHz, CDCl_3_) δ (ppm): 144.37, 143.48, 142.28, 140.26, 139.75, 138.58, 134.69, 130.32, 126.91, 118.09, 111.02, 95.82, 88.83, 31.93, 31.70, 30.75, 29.67, 29.65, 29.59, 29.57, 29.54, 29.47, 29.36, 28.21, 28.19, 22.71, 14.13, −8.04 (Appendix A).

#### 2.1.6. Synthesis of PDTBDT-TE-DTNT

DTBDT-TESn (182 mg, 0.13 mmol), DTNTC_8_Br (103 mg, 0.13 mmol), 10 mL of toluene and 2 mL of *N*,*N*-dimethylformide (DMF) were added to a 25 mL two-neck bottle. After being flushed with argon for 20 min, the catalyst Pd_2_(dba)_3_ (1.5 mg) and P(o-tol)_3_ (3 mg) were added, and the mixture was then purged with argon for 10 min. The solution was stirred and heated to reflux for 48 h under argon atmosphere. At the end of polymerization, the polymer was end-capped with 2-tributylstannylthiophene and 2-bromothiophene to remove bromo and trimethylstannyl end groups. The reaction was cooled to r.t., and the mixture was precipitated in methanol and filtered. Further purification was carried out by Soxhlet extraction using the sequence ethanol, acetone, hexane and toluene as the eluents to remove the residue catalyst and oligomers. Following that, the concentrated solutions of the copolymers in toluene were poured into methanol again (300 mL). The precipitation was collected and dried under vacuum overnight. Yield: 81%. *M*_n_ = 21,628 g/mol with PDI of 3.14.

## 3. Results and Discussion

### 3.1. Synthesis and Characterization of the Monomers and Copolymers

Scheme 1 shows the synthetic routes to the monomers and the copolymer. The 2,7- bis(trimethylstannyl)-5,10-di(4,5-didecylthien-2-yl-ethynyl)dithieno[2,3-*d*:2′,3′-*d*′]benzo[1,2-*b*:4,5-*b*′] dithiophene (DTBDT-TESn) was prepared using the following process: the 4,5-decylthiphene was iodinated with *N*-iodosuccinimide (NIS) to lead 2-iodo-4,5-didecylthiophene (T-I) in ambient conditions under dark, followed by the treatment of 2-trimethylsilyl-acetylene under palladium chloride and cuprous iodide as catalyst to generate the 2-trimethylsilylethynyl-4,5-didecylthiophene (TE-Si). After that, the resultant TE-Si was decomposed in a solution of potassium carbonate of methanol to give 2-(4,5-didecylthien-2-yl)acetylene, and subsequently 2-(4,5-didecylthien-2-yl)ethyeyl lithium and dithieno[2,3-*d*:2′,3′-*d*′] benzo[1,2-*b*:4,5-*b*′]dithiophene-5,10-diketone were reacted in anhydrous THF to get DTBDT-TE. The DTBDT-TE was firstly reacted with n-BuLi in anhydrous THF at −35 °C, then the solution of trimethylstannyl chloride was reacted in anhydrous THF to result in DTBDT-TESn. The structures of the monomers were confirmed by ^1^H NMR and ^13^C NMR before use (Appendix A). The copolymer of PDTBDT-TE-DTNT was synthesized between DTNTC_8_Br and DTBDT-TESn by Stille cross-coupling reaction (Scheme 1) and worked up following the reported procedure with a yield of 81% [28,31,32]. The number-average molecular weight (*M*_n_) of the PDTBDT-TE-DTNT, determined by gel-permeation chromatography (GPC) in THF with polystyrene as standards, was about 21.6 KDa with a PDI of 3.14. The decomposed temperature (*T*_d_, 5% weight-loss) of the PDTBDT-TE-DTNT was about 444 °C under N_2_ flow, indicating that the copolymer exhibited good thermal stability (Appendix A). Meanwhile, the alternating CPs named PDTBDT-T-DTNT, which were derived from DTBDT-TSn and DTNTC_8_Br, were also prepared following the procedures in the references for comparative investigation [25].

### 3.2. Absorption and Aggregation Characteristics of the Copolymers

#### 3.2.1. Absorption Spectra of the Copolymer

The UV-Vis absorption spectra of the copolymer in solution and solid thin film were monitored on a UV-1800 spectrophotometer, and the absorption spectra and corresponding parameters are presented in Figure 1 and Table 1. The PDTBDT-TE-DTNT exhibited three absorption peaks at 423 nm, 545 nm and 643 nm, with shoulder absorption peaks at 699 nm. The absorption peaks at around 423 nm were attributed to the π−π∗ transition of the polymer backbone, and the absorption peaks at around 643 nm (0-1) and 699 nm (0-0) were respectively attributed to the intermolecular charge transfer (ICT) of the polymer [32,33]. Continuing on from solution to solid state, the absorption peak of PDTBDT-TE-DTNT at around 423 nm decreased, the absorption peak at round 643 nm clearly increased, and the on-set band gap wavelength varied from 789 nm to 792 nm. Additionally, the absorption spectra of the controlled polymer, e.g., PDTBDT-T-DTNT, were also monitored for comparison (Appendix A). The PDTBDT-T-DTNT exhibited three main absorption peaks at 347 nm, 484 nm and 673 nm, with a shoulder absorption peak at around 726 nm in dilute solution. The absorption spectrum of PDTBDT-T-DTNT was almost unchanged, except that the relative absorption intensity at 347 nm was obviously decreased, and the on-set band gap wavelength of the PDTBDT-T-DTNT was red-shifted from 809 nm to 822 nm, when moving from solution to solid state. The corresponding optical band gap values ($E_{g}^{\mathrm{opt}}$) of PDTBDT-TE-DTNT and PDTBDT-T-DTNT were 1.56 and 1.51 eV, respectively, according to the formula $E_{g}^{\mathrm{opt}}$ = 1240/$\lambda_{\mathrm{onset}}^{\mathrm{film}}.$ It was found that PDTBDT-TE-DTNT presented a slightly broader optical band gap in contrast to that of PDTBDT-T-DTNT.

#### 3.2.2. Aggregation of the Copolymers in Solution and Solid State

To investigate the aggregation characteristics of the CPs in solution, temperature-dependent absorption spectra (TD-Abs) of the copolymers in o-dichlorobenzene (o-DCB) solution were monitored (Figure 2). Obvious changing of the absorption spectra of PDTBDT-TE-DTNT and PDTBDT-T-DTNT in o-DCB solution was found during the heating process. From 25 to 105 °C, the absorbance of the λ_0-0_ peaks at 699 nm and the λ_0-1_ peaks at 643 nm for PDTBDT-TE-DTNT gradually decreased and blue-shifted by 20 and 32 nm, respectively, indicating that the degree of torsion of the unit on the polymer conjugated backbone was aggravated, the coplanarity of the molecule was destroyed, and the effective conjugate length and the degree of conjugation were reduced, thus causing a blue shift in the absorption spectrum [33,34,35,36]. For the PDTBDT-T-DTNT, the absorbance of the λ_0-0_ peaks at 726 nm and the λ_0-1_ peaks at 673 nm was continuously reduced and blue-shifted to 689 nm and 580 nm. In addition, the relative intension rations of the λ_0-1_/λ_0-0_ peaks varied between 0.96:1 and 1.23:1 for PDTBDT-TE-DTNT and 1.05:1 and 1.97:1 for PDTBDT-T-DTNT. It was clear that the blue shift value of λ_0-1_ peak and the decrease of λ_0-0_ peak for PDTBDT-TE-DTNT were significantly smaller than those for PDTBDT-T-DTNT, and the results indicated that PDTBDT-TE-DTNT exhibited stronger aggregation than PDTBDT-T-DTNT in dilute solution [28,33,34,35,36].

To gain insight into the aggregation characteristics of CPs in solid state, X-ray diffraction (XRD) analyses of the polymer films cast from CB solution onto glass substrate were measured. As shown in Figure 3, The diffraction peaks of PDTBDT-TE-DTNT were located at 2θ = 22.31° and 2θ = 3.31°, corresponding to the π–π stacking d-spacing of 3.98 Å and interlayer d-spacing of 26.66 Å based on Bragg’s law (i.e., λ = 2dsinθ) [28]. The controlled polymer PDTBDT-T-DTNT films provided two peaks at about 2θ = 3.27° and 2θ = 23.01°, corresponding to an interlayer d-spacing of 26.99 Å and a d-spacing of 3.86 Å, respectively. It was shown that PDTBDT-TE-DTNT possessed a shorter interlayer d-spacing distance and a longer π-π stacking distance than PDTBDT-T-DTNT.

### 3.3. Electrochemical Characteristic of the Copolymers

The energy levels are important parameters to guide the selection of appropriate acceptors in PVCs. Therefore, cyclic voltammetry (CV) was employed to measure the redox behaviors of the copolymers and determine their energy levels [37,38]. The onset oxidation potential was determined by the CV curves and calibrated with the potential of ferrocene/ferrocenium (Fc/Fc^+^), assuming the energy level of ferrocene/ferrocenium (Fc/Fc^+^) to be −4.80 eV below the vacuum level [39]. The redox potential of Fc/Fc^+^ under the above-mentioned conditions was +0.11 V. For the CV of ferrocene and the two polymer films on a glassy carbon electrode in 0.1 mol L^−1^ tetrabutylammonium hexafluorophosphate (Bu_4_NPF_6_) acetonitrile solution at a scan rate of 50 mV s^−1^, the HOMO levels and LUMO levels were calculated by empirical formulas (*E*_HOMO_ = −(Eox + 4.69) (eV) [39] and *E*_LUMO_ = −(│*E*_HOMO_│−$\;E_{g})\;$(eV). As shown in Figure 4a, the onset oxidation potential of PDTBDT-TE-DTNT was located at around 0.68 V, corresponding to the HOMO level of −5.23 eV, and the LUMO energy level was about −3.67 eV. Meanwhile, the HOMO and LUMO levels of PDTBDT-T-DTNT were about −5.32 eV and −3.81 eV, respectively. The elevated HOMO energy level of PDTBDT-TE-DTNT would result in a decrease of the *V*_OC_ of the PDTBDT-TE-DTNT-based PVCs in comparison to that for the PDTBDT-T-DTNT-based PVCs [40]. In spite that, the LUMO gaps between the copolymers and PC_71_BM were 0.26–0.39 eV, which would provide sufficient driving force to promote efficient exciton dissociation at the D–A interface, thereby ensuring energetically favorable electron transfer between the CPs and the PC_71_BM [41].

### 3.4. Hole Mobilities of the Blend Films from the Copolymers

The carrier charge mobility is an important parameter, and could affect charge carrier transport and recombination in photovoltaic devices. Higher mobility is beneficial to increasing the current density and decreasing unfavorable exciton recombination [42,43]. The space-charge-limited current (SCLC) method was used to determine the hole mobilities of the blend films from the polymer in the devices with configuration of ITO/PEDOT:PSS/polymer: PC_71_BM/MoO_3_/Ag. The mobilities of the blend films were calculated by the SCLC model, which is described by Equation (1) [44]

(1)$$\mu =\frac{8L^{3}J}{9\varepsilon_{0}\varepsilon_{r}V^{2}}$$ where *J* is current density, ε_0_ stands for the permittivity of free space, ε_r_ is the relative dielectric constant of the transport medium, which is assumed to be around 3 for the CPs, µ is the hole mobility, *V* is the internal potential in the devices and *L* is the thickness of the active layers. The *J*-*V* characteristics of the devices from the PDTBDT-TE-DTNT/PC_71_BM were presented in Figure 5, and the *J*-*V* curve of the devices from PDTBDT-T-DTNT/PC_71_BM is also provided for comparison. The hole mobilities for PDTBDT-TE-DTNT/PC_71_BM was about 1.85 × 10^−4^ cm^2^·V^−1^·s^−1^, which is comparable to the value of 1.55 × 10^−4^ cm^2^·V^−1^·s^−1^ obtained for the PDTBDT-T-DTNT/PC_71_BM blend films. In addition, the results indicated that the replacement of T with TE did not influenced the hole mobilities of the copolymers.

### 3.5. Photovoltaic Characteristics and Optical Modeling of the Device from the Copolymers

To explore the influence of the flanks on the photovoltaic properties of the copolymers of the PDTBDT-TE-DTNT and PDTBDT-T-DTNT, inverted photovoltaic devices (i-PVCs) with a configuration of ITO/PFN/active layer/MoO_3_/Ag, in which PDTBDT-TE-DTNT and/or PDTBDT-T- DTNT was used as electron donor material, and PC_71_BM was used as electron acceptor material, were fabricated following the reported procedures [25]. The current density vs voltage (*J*-*V*) characteristics of i-PVCs are shown in Figure 6a, as measured under illumination of AM 1.5G at 100 mW/cm^2^ conditions, and the corresponding photovoltaic performances are listed in Table 2. The optimal weight ratios of the PDTBDT-TE-DTNT and PC_71_BM were about 1:1, while the weight ratios of the PDTBDT-TE-DTNT and PC_61_BM were varied from 1:1, 1:2 and 1:3, 3% DIO was employed as solvent additives (Appendix A and Appendix A). The PCEs of the optimal i-PVCs from PDTBDT-TE-DTNT/PC_71_BM (W:W, 1:1) with 3% DIO as solvent additives, were about 3.97%, alongside with a *V*_OC_ of 0.60 V, a *J*_SC_ of 10.15 mA cm^–2^ and a FF of 65.16%. Meanwhile, i-PVCs from controlled CPs of PDTBDT-T-DTNT/PC_71_BM provided the maximal PCE of 7.57% with a *V*_OC_ of 0.70 V, a *J*_SC_ of 16.09 mA/cm^2^ and a FF of 67.19%. It could be found that the *V*_OC_ and *J*_SC_ of i-PVCs from PDTBDT-TE-DTNT/PC_71_BM blend films were lower than those for i-PVCs from the PDTBDT-T-DTNT/PC_71_BM. The lower *V*_OC_ of PDTBDT-TE-DTNT-based devices might be attributed to the enhanced HOMO energy level, and the broader band gap of PDTBDT-TE-DTNT might lead to the decreasing *J*_SC_ for PDTBDT-TE-DTNT-based devices in contrast to that for PDTBDT-T-DTNT. However, it is noted that the estimated *J*_SC_ of the devices from the copolymers were calculated under the assumption that the devices exhibited the same internal quantum efficiencies, charge transporting and collection characteristics, etc., except for the difference in light harvesting between them (Appendix A), which were about 23.2 and 21.8 mA/cm^2^. The results indicated that there should be another reason behind the lower *J*_SC_ of the PDTBDT-TE-DTNT-based devices in comparison to the PDTBDT-T-DTNT-based devices, in addition to the broadening of the bandgap of PDTBDT-TE-DTNT.

Recently, it has been well demonstrated that the topography and morphology of the active layer (such as domain size) are critical to exciton separation, and charge carrier recombination and transport, which also play an important role in *J*_SC_. Tapping-mode atomic force microscopy (AFM) and transmission electron microscopy (TEM) were implemented to gain insight into the nature of topography and morphology of the blend films from the copolymers. As demonstrated by the AFM height images in Appendix A, the root-mean-square surface roughness (RMS) values were about 1.64 nm for PDTBDT-T-DTNT/PC_71_BM and 2.66 nm for PDTBDT-TE-DTNT/PC_71_BM blend films. The TEM images of the corresponding films indicated that that the PDTBDT-T-DTNT/PC_71_BM exhibited similar domain sizes to the PC_71_BM (dark regions) and the polymers matrix (bight regions) with PDTBDT-TE-DTNT/PC_71_BM blend films (Appendix A) [45,46,47,48]. It is worthwhile pointing out that the results of the study on the relationship between RMS of the blend films and *J*_SC_ for the PVCs are, thus far, contradictory [49,50]. However, we speculated that the lower *J*_SC_ of the PDTBDT-TE-DTNT-based i-PVCs cannot be ascribed to the slight difference in RMS between the PDTBDT-T-DTNT/PC_71_BM and PDTBDT-TE-DTNT, because the blend films with the PDTBDT-TE-DTNT and PDTBDT-T-DTNT exhibited similar morphologies [45,46,47,48,49,50].

To gain insight into the origin of the large decrease in *J*_SC_ of i-PVCs from PDTBDT-TE-DTNT, we also monitored the incident photon-to-electron conversion efficiencies (IPCEs) of i-PVCs from the copolymers (Figure 6b). The PDTBDT-TE-DTNT/PC_71_BM-based devices presented light response ranging from 300 nm to 800 nm with IPCEs of 1.2% to 48.8%. As compared with the PDTBDT-TE-DTNT-based devices, the PDTBDT-T-DTNT/PC_71_BM-based devices presented similar light response except that they exhibited higher IPCEs in the whole photocurrent response of 300 nm to 820 nm. In addition, the lower *J*_SC_ of i-PVCs from PDTBDT-TE-DTNT might have led to the lower IPCEs of the devices in contrast with those for the devices from PDTBDT-T-DTNT. Following that, the characteristics of the photocurrent density (*J_ph_*) versus the effective applied voltage (*V_eff_*) of the i-PVCs were also measured [51,52]. In addition, the *J_ph_* is defined as per Equation (2)

(2)$$J_{ph}\;=J_{L}-J_{D}$$ where *J_L_* and *J_D_* are the photocurrent densities under illumination and dark, respectively. *V_eff_* can be defined as the difference between *V_a_* and *V*_0_ (3)

(3)$$V_{eff}=V_{a}\;-V_{0}$$ where *V*_0_ is the voltage at which the photocurrent is zero and *V_a_* is the applied external voltage bias. As shown in Figure 7, the *J_ph_* of the both the devices from PDTBDT-TE-DTNT and PDTBDT-T-DTNT increased sharply when *V_eff_* was low, and then saturated when the *V_eff_* increased, suggesting that most of the excitons had been dissociated with the help of a large reverse bias. Nevertheless, it could be found that the *J_ph_* of the PDTBDT-T-DTNT-based devices were always higher than those for PDTBDT-TE-DTNT. In addition, the ratio of *J_ph_* to *J*_sat_ under short-circuit condition was defined as *P*_diss_ to present the exciton dissociation probability, and in which *J*_sat_ was the saturated current density. *P*_diss_ of about 91% and 85% were obtained for the optimal i-PVCs from PDTBDT-T-DTNT and/or PDTBDT-TE-DTNT, respectively. The results indicated that the PDTBDT-TE-DTNT-based i-PVCs exhibited low *P*_diss_, which would result in lower IPCEs for the PDTBDT-TE-DTNT-based i-PVCs.

### 3.6. Time-Resolved Photoluminescence of the Copolymers

Generally, polymer singlet excitons are the precursors for charger photogeneration taking place at the D–A interface, and the lifetime of the singlet excitons in CPs would determine their diffusion length in polymer films, which would exhibit a direct impact on the exciton dissociation and photocurrent generation characteristics of the photovoltaic devices [53,54]. The time-resolved photoluminescence (PL) of the PDTBDT-TE-DTNT and PDTBDT-T-DTNT was performed by time-correlated single photon counting (TCSPC) on the dilute solution of toluene (Figure 8). The τ_0_ of 2.61 ns for PDTBDT-T-DTNT, and 1.92 ns for PDTBDT-TE-DTNT, was extracted from the time-resolved photoluminescence decay for the copolymer via single exponential fit. However, it could be found that the single exponential fit failed to accurately describe the PL decay of PDTBDT-TE-DTNT, and the deviations of PDTBDT-T-DTNT PL decay from the single exponential fit were very small. Double exponential fits were also implemented to understand the PL decay of the copolymers. The τ_1_ and τ_2_ of 0.347 and 2.741 ns were extracted from the PL decay of PDTBDT-T-DTNT, and the τ_1_ and τ_2_ of 0.486 ns and 2.17 ns were extracted from the PL decay of PDTBDT-TE-DTNT. As compared with the single exponential fit, the double exponential fits could more accurately describe the PL decay for both copolymers. This indicates that there were at least two types of excited state decay for the excitations of the copolymers, and the dominant excited states, which were mainly related with the charger photo-generation process of the photovoltaic devices, provided a lifetime of 2.741 ns for PDTBDT-T-DTNT and 2.171 ns for PDTBDT-TE-DTNT. In a word, the PDTBDT-TE-DTNT had a shorter PL lifetime, and the lower exciton dissociation probability of the devices might be attributed to the shorter PL lifetime when compared to that of PDTBDT-T-DTNT.

### 3.7. Computational Consideration of the Copolymers

Assuming that the only difference between the two polymers is that the donor units have different flanks, it makes sense that discrepancy between the optoelectronic and aggregation, etc., of the copolymers would mainly result from their different flanks. In addition, computational calculations were implemented by using density functional theory (DFT) calculations with the B3LYP/6-31G (d,p) basis set in Gaussian 09, striking a balance between prediction of the conformation and a completion of the calculations within a reasonable time, and the alkyl side groups were replaced with methyl [55]. The trans- (or cis-) coplanar conformations of the DTBDT-T and DTBDT-TE were respectively defined by the dihedral angles (torsion angels) between the flanks of dialkylthiophene in T or TE and DTBDT planar core (Figure 9). The optimized ground-state geometries of the DTBDT-T-DTNT and DTBDT-TE-DTNT systems were presented in Figure 9a, in which the dihedral angles were about 75.5° and 1.5°, respectively. Meanwhile, the relaxed potential-energy scans of the T in DTBDT-T-DTNT and/or TE in DTBDT-TE-DTNT were implemented, and the results signified that the TE flanks in DTBDT-TE-DTNT exhibited narrower potential energy wells than that for the T flanks in DTBDT-T-DTNT, and indicated that the TE flanks in the DTBDT-TE-DTNT presented lower rotating energy barrier near the minimum energy conformation in contrast to T in the DTBDT-T-DTNT, which might lead to easier rotation of the TE flanks, and the enhancement of the energy relaxation channels of the excited states [56], thus contributing to the shorter PL lifetime and the lower exciton dissociation probability of the PDTBDT-TE-DTNT and/or PDTBDT-TE-DTNT-based PVCs compared with those for PDTBDT-T-DTNT and/or PDTBDT-T-DTNT-based PVCs (Figure 7 and Figure 8). 

## 4. Conclusions 

In conclusion, we synthesized an alternating conjugated polymer, named as PDTBDT-TE-DTNT, from DTBDT-TE and DTNT, and the controlled polymer PDTBDT-T-DTNT was also prepared for comparison. Their optimal *i-*PVCs provided PCEs of 3.97% and 7.57%, with a *V*_OC_ of 0.60 V and 0.70 V, *J*_SC_ of 10.15 mA/cm^2^ and 16.09 mA/cm^2^ and *FF* of 65.16% and 67.19% under 100 mW/cm^2^ illumination (AM 1.5G), respectively. Subsequently, comparative investigation of the absorption, aggregation, charge transporting, photovoltaic characteristics, etc., of PDTBDT-TE-DTNT and PDTBDT-T-DTNT were implemented. It was found that that the replacement of T with TE flanks would lead to a slight broadening of the band gap and elevation of the HOMO energy levels of the PDTBDT-TE-DTNT relative to PDTBDT-T-DTNT. Moreover, the optical calculations, exciton dissociation probability, time-resolved photoluminescence measurements and computational considerations results revealed that the lower torsional barrier of the TE flanks might contribute to the enhancement of the relaxation patterns of the excitons in the PDTBDT-TE-DTNT, thus leading to a decrease in the photo-induced excited state lifetime and exciton dissociation probability, which could be mainly attributed to the lower *J*_SC_ of the PVCs from PDTBDT-TE-DTNT, in contrast to those for PDTBDT-T-DTNT and PDTBDT-T-DTNT-based PVCs.

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
