# Peer review of "Effect of Flank Rotation on the Photovoltaic Properties of Dithieno[2,3-d:2′,3′-d′]benzo[1,2-b:4,5-b′]dithiophene-Based Narrow Band Gap Copolymers"

_polymers, 2019, doi:10.3390/polym11020239_

Round 1
Reviewer 1 Report
Summary: in this manuscript, Zhang et.al. investigated the effect of aromatic side chains on the properties of conjugated polymers and the photovoltaic devices. Compared to the control polymer, the novel polymer incorporated an ethynyl unit between the backbone and the aromatic side unit. The polymers and the photovoltaic device weres well characterized. I would recommend publishing this manuscript on Polymers with some minor issues revised.
Minor issue
The author thought the lower rotation barrier induced by the triple bond caused shorter exciton lifetime, smaller exciton dissociation possibility and thus lower JSC. However, it is recommended to investigate the morphology of the photovoltaic devices because in addition to the polymer structure, the morphology of the active layer (such as domain size) are critical to the exciton separation, charge carrier recombination and transport, which also play an important role in Jsc.
Author Response
Response 1: We have described the morphology, AFM and TEM measurements were carried out to study the surface morphology of the PDTBDT-TE-DTNT/PC71BM and PDTBDT-T-DTNT/PC71BM blend layers in Figure S7 and Figure S8. And the corresponding description has been added in the revised manuscript as following:
“In recently, it has been well demonstrated that the topography and morphology of the active layer (such as domain size) are critical to the exciton separation, charge carrier recombination and transport, which also play an important role in JSC. Tapping-mode atomic force microscopy (AFM) and transmission electron microscopy (TEM) were implemented to gain insight into the nature of topography and morphology of the blend films from the copolymers. As demonstrated by the AFM height images in Figure S7, the root-mean-square surface roughness (RMS) value were about 1.64 nm for PDTBDT-T-DTNT/PC71BM and 2.66 nm for PDTBDT-TE-DTNT/PC71BM blend films. The TEM images of the corresponding films indicated that that the PDTBDT-T-DTNT/PC71BM exhibited similar domain sizes of the PC71BM (dark regions) and polymers matrix (bight regions) with PDTBDT-TE-DTNT/PC71BM blend films (Figure S8) [45–48]. It is worthwhile pointing out that the results of the study on the relationship between RMS of the blend films and JSC for the PVCs are contradictory so far [49,50]. However, we speculated that lower JSC of the PDTBDT-TE-DTNT based i-PVCs can not ascribed to slightly difference RMS between the PDTBDT-T-DTNT/PC71BM and PDTBDT-TE-DTNT, while the blend films with the PDTBDT-TE-DTNT and PDTBDT-T-DTNT held similar morphologies [45–50]. ”

Reviewer 2 Report
The authors deal with the synthesis and characterization of some new copolymers to be employed as photoactive layers for PVCs solar cells. The paper is clear, well arranged and the experiments well described. However, some points have to be better explained.
page 9 line 324 Why an inverted structure of the cell has been employed instead of the conventional architecture?
line 330 The authors have some informations about the electrical behavior of the cells with other PCBM/polymer ratios?
line 336-338 this sentence is not clear
the english should be improved
Author Response
Response 1: Compared with conventional PVCs, inverted type devices demonstrate better long-term ambient stability by avoiding the need for the corrosive and hygroscopic hole-transporting poly(3,4-ethylenedioxylenethiophene):poly(styrenesu-lphonic acid) (PEDOT:PSS) and low-work-function metal cathode, both of which are detrimental to device lifetime (Hsieh, C.-H. Highly efficient and stable inverted polymer solar cells integrated with a cross-linked fullerene material as an interlayer. J. Am. Chem. Soc. 2010, 132, 4887—4893; Xu, Z. Vertical phase separation in poly(3-hexylthiophene): fullerene derivative blends and its advantage for inverted structure solar cells. Adv. Funct. Mater. 2009, 19, 1227—1234.). Moreover, it has been demonstrated that the inverted PVCs may take advantages such as the vertical phase separation, reduction of bimolecular recombination, and enhancement of absorption of photons etc., thus to improve the Jsc and FF of the inverted PVCs. The inverted device structure is therefore an ideal configuration for all types of PVCs (He, Z.; Zhong, C.; Su, S.; Xu, M.; Wu, H.; Cao,Y. Enhanced power-conversion efficiency in polymer solar cells using an inverted device structure. Nat. Photonics 2012, 6, 591—595.).
Response 2: The optimal weight ratios of the PDTBDT-TE-DTNT and PC71BM were about 1:1, while the weight ratios of the PDTBDT-TE-DTNT and PC61BM were varied from 1:1, 1:2 and 1:3, 3% DIO was employed as solvent additives (Figure S5 and Table S1).
Response 3: The sentence “The lower VOC and JSC of PDTBDT-TE-DTNT based devices might be contributed to the enhanced HOMO energy level and broader band gap of PDTBDT-TE-DTNT in contrast to those for PDTBDT-T-DTNT.” was modified as “ The lower VOC of PDTBDT-TE-DTNT based devices might be attributed to the enhanced HOMO energy level and the broader band gap of PDTBDT-TE-DTNT might led to the decreasing JSC for PDTBDT-TE-DTNT based devices in contrast to those for PDTBDT-T-DTNT.”
